# Risk Factors Regarding Dog Euthanasia and Causes of Death at a Veterinary Teaching Hospital in Italy: Preliminary Results

**DOI:** 10.3390/vetsci9100554

**Published:** 2022-10-09

**Authors:** Michela Pugliese, Annastella Falcone, Angela Alibrandi, Agata Zirilli, Annamaria Passantino

**Affiliations:** 1Department of Veterinary Sciences, University of Messina, 98122 Messina, Italy; 2Unit of Statistical and Mathematical Sciences, Department of Economics, University of Messina, 98122 Messina, Italy

**Keywords:** euthanasia, unassisted death, dog, risk factors

## Abstract

**Simple Summary:**

The current study aimed to report proportional euthanasia and unassisted death, as well as to identify demographic and clinic risk factors associated with the main clinical causes and method of death (euthanasia and unassisted) in a population of dogs retrieved from an Italian referral veterinary teaching hospital. Deaths by euthanasia were estimated to have a prevalence of 40.7%, by dying unassisted 50.8%, whilst only 8.5% of dogs died accidentally. The death of euthanasia was primarily due to neoplastic, degenerative, and congenital diseases, and female gender, age, and neoplastic and degenerative processes were considered significant risk predictors. In unassisted deaths, the risk predictors of male gender, age, and infection/inflammatory conditions were considered significant.

**Abstract:**

The decision to request and proceed with euthanasia in a dog is complex and predictors of such decisions are important. This study investigates the risk factors (demographic and clinical) associated with the main clinical causes and methods of death (euthanasia or unassisted death) in a population of dogs. By comparing euthanasia to unassisted deaths, the authors assess causes of death to evaluate their relative impacts on decision-making to choose euthanasia compared with an unassisted death. For this, goal data from electronic medical records of dogs who had died (unassisted death and euthanasia), obtained from an Italian referral veterinary teaching hospital from 2010 to 2020, were analyzed. The causes of death were categorized by pathophysiological process and the organ system. Univariate and multivariable logistic regression analyses were performed to identify the factors that significantly affect the probability of undergoing euthanasia and to individuate independent significant predictors of euthanasia and unassisted death, respectively. Death rate by euthanasia was 40.7% (125/307), by died unassisted 50.8% (156/307), whilst only 8.5% of dogs (26/307) died accidentally. The main causes of death for euthanasia were due to neoplastic (75.6%), degenerative (64.3%), and congenital (60%) diseases. Furthermore, the findings reveal that in deaths by euthanasia, the significant risk predictors were female gender, age, and neoplastic and degenerative processes; while in unassisted deaths, the significant risk predictors were male gender, age, and infection/inflammatory conditions. These preliminary outcomes highlight the information of this study which may be used to evaluate strategic interventions and health promotion strategies to be implemented, with consequent welfare gains for the canine population.

## 1. Introduction

The increasing importance of chronic diseases, neoplastic diseases, and traumatic events in companion animals (CAs) as a cause of death [1] has created interest in the role of veterinary medicine in the timing and mode of death and dying.

In many instances, death is not merely the result of the natural course of a lethal disease; medical decision-making often contributes [1]. Depending on the case, veterinarians may recommend euthanasia based on the animal’s poor quality of life, or in patients with a negative prognosis who suffer pain due to severe illness or injury. The greatest ethical meaning is attributed to the quality rather than the quantity of the animals’ life. The decision to euthanize an animal can be done with rationality based on compassion, convenience, or economic considerations, or can be applied to the principle of best interest that considers what is best for the animal regardless of the owner’s wishes. All these aspects are important to take into consideration when discussing the management and health-related factors associated with euthanasia. To assist the owner in the euthanasia decision-making process, it could be useful to apply the algorithm for euthanasia decisions produced by the BVAs (British Veterinary Association) Ethics and Welfare Group [2]. The scale of Villalobos [3] also represents a straightforward means for veterinarians and owners to assess dogs’ quality of life. This tool would help to monitor, objectively, the progression of the animal’s condition and agree on whether euthanasia is necessary and ethically appropriate.

Difficult situations can occur when owners feel that the suffering of their animals is unbearable, feel hopeless, and ask for help from the veterinarian to terminate the life of their dogs through euthanasia [4,5,6,7,8]. It is acceptable and sometimes deemed necessary when an animal is suffering due to an incurable illness or injury, or when an animal presents a significant risk to human health and safety or the safety of other animals, through disease or aggressive behavior [6,7,8,9]. So, the veterinarian often must balance the interests of a chronically ill, aged, or unwanted animal with those of the owner, who may wish for treatment to be continued or stopped against veterinary advice.

The choice to euthanize could be rationally made based on economic, convenience, or compassionate reasons [10]. Gorodetsky (1997) [11] states that, generally, old age is the most common reason for euthanasia, followed by terminal sickness, aggression, and behavioral alterations [7,8]. Nevertheless, the decision to euthanize a dog is a complex topic because it intersects with important emotional and societal values, because the dog deserves consideration as a sentient being [12].

Facing a dog’s death, and especially euthanasia, could be a potential moral distress also for the veterinary team [4,13]. Veterinarians might feel reluctant to provide euthanasia as an option, viewing it as an admission of failure of their treatment skills [14], or because the animal may be suffering from a treatable condition, but the owner may not be able to afford the proposed treatment for financial reasons [10], indicated as economic euthanasia [12]. The owner has the right to do it because animals are considered to be personal property [12]. The animal patient is the owner’s property and is usually considered not able to participate in the decision-making [6].

In Italy, laws that establish rules for pets’ euthanasia are not present. The conditions under which euthanasia of dogs and cats is justified are only partly regulated by Law no. 281/91 that, besides delegating the work of birth control in cat and dog populations to the regions, has made it statutory that stray dogs may only be euthanized when they are “seriously or incurably ill or proved to be dangerous”.

Studies on methods of death and the risk of euthanasia in shelters and owned dogs have been carried out, particularly in the US, Canada, and the UK [15,16,17,18,19,20]. Canine mortality is described as either due to unassisted death or euthanasia [20]. In previous research, neoplastic diseases, musculoskeletal disorders, and neurological disorders have been reported as the most common reasons for mortality in dogs [21]. Another research reported the main reasons for euthanasia and causes of unassisted death in dogs admitted to a Brazilian university teaching hospital [22]. Similar studies have not been previously carried out in Italy, mostly due to the lack of information regarding the number and characteristics of the canine population, as well as the number of canine euthanasia, highlighting the need for data. Considering this limited evidence, the aims of the current retrospective analysis of clinic data retrieved from an anonymized Italian referral veterinary teaching hospital were to (1) report proportional unassisted death (when the animal patient died upon being presented to the hospital or while attempting/confirming diagnosis and/or treatment) and euthanasia; and to (2) identify the demographic and clinic risk factors associated with the main causes and method of death (unassisted or euthanasia).

## 2. Materials and Methods

### 2.1. Data Collection and Study Population

Anonymized clinical data (deidentified after selection) were retrieved retrospectively from a search of the electronic medical records database of canine cases presented between 1 January 2010 and 31 December 2020 from a small animal veterinary referral hospital. All dogs with an outcome reported as dead (unassisted and/or accidental) or euthanized were initially identified as the cohort used for the analysis. 

The information retrieved from the records included the breed, date of birth, gender (male/female), neuter status, method/manner of death [whether the death was an unassisted (UD) or accidental death (AD) (death involved in a road traffic accident), or euthanasia (E)], and physio-pathological process (PP) affecting the animals at the moment of diagnosis. 

The registered causes of death (understood as the main or first reported clinical cause for the death) were grouped into eight categories: infectious/inflammatory conditions, congenital diseases, degenerative diseases, metabolic disorders, vascular diseases, toxics, neoplastic processes, and traumatic events. 

All death causes took also into consideration the organ system (OS). The OS categories were cardiovascular, respiratory, gastrointestinal, hepatic, renal/urinary, neurologic, musculoskeletal, hematopoietic, dermatological, and systemic/multiorgan. 

Dogs for which the sole diagnosis was “death due to unknown” were identified and excluded from the analysis.

### 2.2. Statistical Analysis

The categorical variables (breed, gender (male/female), neuter status, method of death, and PP) were expressed as absolute frequencies and percentages and the numerical parameter (age in months) was expressed as the mean, standard deviation, and quartiles (Q1, Q2 and Q3).

Univariate logistic regression models were estimated to identify the factors that significantly affect the probability of undergoing E [23]. A multivariate logistic model was also utilized to individuate the independent significant predictors of E. In both approaches, the tested covariates were age, gender, purebred (yes or no), sterilization (yes or no), owned (yes or no), infectious or inflammatory conditions (yes or no), neoplastic processes (yes or no), traumatic events (yes or no), degenerative process (yes or no), and toxic process (yes or no). 

Congenital, metabolic, and vascular pathophysiological processes were not included in the model due to the small number of registered cases (less than 5).

All tested variables in the univariate models were then inserted into the multivariate model to evaluate the global explanatory power of all covariates.

The same analyses were performed to individuate the significant predictors of UD.

The results of the univariate and multivariate logistic regression models were expressed as an odds ratio (OR), a 95% confidence interval (95% C.I.), and *p*-value. 

A *p*-value < 0.05 was considered statistically significant.

Statistical Analysis was performed by using SPSS for Windows Package, version 22.0.

## 3. Results

### 3.1. Descriptive Statistics of the Study Population

From the initial combined dataset of 578 dogs, 271 were removed because of missing data and they were excluded from the analysis (Figure 1). In total, only 307 animals met the eligibility criteria. Out of the animals, 44.71 % (*n* = 137) were female and 55.4% (*n* = 170) were male, 16.3 % (*n* = 50) were neutered, 31.3% (*n* = 96) were purebred, and 64.5% (*n* = 198) were owned. At the time of death/euthanasia, the median age was 96 months (IQR 48–144 months).

### 3.2. Rate and Type of Death

Relating to information on the recorded death method, 125 dogs (40.7%) were euthanized and 156 (50.8%) died unassisted, whilst only 26 (8.5%) died accidentally (Figure 1).

Table 1 shows the absolute frequencies and percentages calculated for the method of the deaths in relation to the PP and OS involved. It also reports the *p*-values (*p*) referred to as the comparisons between “Euthanasia” and “Unassisted death” in relation to the PP and OS. The same comparisons were not made for “Accidental death” due to the small number of cases.

The most frequent causes of death were inflammatory/infectious processes (152; 49.5%), traumatic events (55; 17.9%), and neoplasia (45; 14.37%), but E was mainly carried out in neoplastic (34; 75.6%), degenerative (18; 64.3%), and congenital (3; 60%) diseases. The OS most involved in cases of E were neurological (31; 75.6%), hepatic (3; 75%), and hematopoietic (5; 62.5%).

### 3.3. Risk Factors

The results of the univariate and multivariate binary logistic regression analysis for the risk factors for E and UD are reported in Table 2 and Table 3, respectively.

The results of the logistic regression model, both univariate and multivariate, showed that there are significant predictors of E (*p* < 0.050), some of which indicated an increased risk (those whose OR was greater than 1), whilst others seemed to exert a “protective” role (OR less than 1).

Specifically, by univariate models, the significant risk predictors were female gender, age, and neoplastic and degenerative processes; on the contrary, significant “protective” factors seemed to be infection/inflammatory conditions and traumatic events (which despite having a significant *p*-value, showed an OR value less than 1).

The multivariate model confirmed the significance already obtained in the univariate models: in particular, female dogs were exposed to a higher risk of E (OR = 1.867; *p* = 0.027), as were older age dogs (OR = 1.01, *p* = 0.001), and those with a neoplastic process (OR = 2.585; *p* = 0.043). Inflammations/infections conditions (OR = 0.481; *p* = 0.041) and traumatic events (OR = 0.230; *p* = 0.002) showed a significant lower exposure of the animal at risk of E.

Relating to the “unassisted death”, the results of the univariate logistic regression model showed that there were significant predictors. The significant risk predictors were male gender, age, and infection/inflammatory conditions; on the contrary, significant “protective” factors of UD seemed to be neoplastic processes and traumatic events (which despite having a significant *p*-value, showed an OR value less than 1).

The multivariate model highlighted most of the significances obtained in the univariate models: specifically, male dogs were exposed to a higher risk of UD (OR= 1.878; *p* = 0.022), as were older age dogs (OR = 1.743, *p* = 0.013), and those with an infection/inflammatory condition (OR = 2.179; *p* = 0.030). Traumatic events (OR= 0.383; *p* = 0.033) showed a significant lower exposure of the animal at risk of UD; finally, the neoplastic processes showed a not significant *p*-value (*p* = 0.055). Therefore, it cannot be considered a significant predictor of UD (unlike what was found for E).

Finally, through data filtering operations, the type of profile of the most exposed dog at risk of E has been outlined, and it was possible to affirm that the condition that exposes them to the greatest risk is the PP of a neoplastic nature; the most exposed to the risk are those which are crossbred, female, with an average age of about 11 years (137.8 months), not sterilized, and are owned dogs. They are euthanized in 75% of cases.

## 4. Discussion

The data from this study provided information on the death rates, method and causes of death, risk predictors, and protective factors in a population of hospitalized dogs in an Italian referral veterinary hospital. 

Within this study, the number of animals that died by UD was higher than those who were euthanized. These were animals hospitalized in an emergency and in a critical condition, and probably the UD occurred before taking the decision to euthanize. Indeed, E was performed only in 40.7% of cases.

Higher euthanasia values are reported in the literature. In a study conducted in the United Kingdom, the frequency of euthanasia was 89.3% [20], 85.2% in the United States [24], 86.4% in England [25], 91% in New Zealand [26], while in Brazil there was 48% [27], and 21.8% in Taiwan [28].

Overall, among the general categories relating to the PPs, inflammatory/infectious conditions had the highest number of animals that died. It is expected that the high mortality rate in this study might have been associated with the lack of vaccinations against infectious diseases, such as canine hepatitis, canine distemper, parvo-enteritis, and parainfluenza. A previous study considered infectious diseases the most common cause of death, especially in puppies [29].

Within each OS category, although there was a systemic involvement (multiorgan) overall, gastrointestinal, renal/urinary, and neurological systems ranked 2nd, 3rd, and 4th, respectively. Infectious diseases of the gastrointestinal system are reported to be the main cause of death in puppies and young dogs [30,31].

The main disease identified was chronic kidney disease (CKD) with complications derived from an advanced stage (stages III and IV), leading the patient to have a shorter survival concerning CKD severity, as described by some authors [32,33,34]. 

Although several data relating to the presumptive diagnoses were missing in medical records, some diseases that involved the neurological system were idiopathic epilepsy and intervertebral disc disease. Only one case of suspected diagnosis of canine distemper encephalitis was identified due, based on clinical history, to the non-vaccination against canine distemper virus. In a retrospective study that reports different causes of diseases affecting the central nervous system in England, similar results are described [35]. Nevertheless, this lack of data could have also biased the understanding of the results, however, it is possible to suppose that owners who were less inclined to give their dogs in-depth medical examinations for the expected poor prognosis or those who could not afford to pay for veterinary care opt for the euthanasia of their dog.

All dogs that died by UD were also critical and dying patients with metabolic disorder, inflammatory/infectious conditions (in 2nd rank), or toxic diseases (in 3rd rank) that had significant care needs for which efforts had been undertaken to stabilize them without any success. Additionally, in the case of traumatic events with serious and often multiple injuries being the cause of death after their presentation, only a few cases required E due to a poor prognosis (18.2%). Among infectious diseases, beyond those mentioned above, given the positivity to specific tests (immunofluorescence antibody test, IFAT, and polymerase chain reaction, PCR) performed on dogs, as reported in medical records, and the high prevalence of canine leishmaniosis in Italy [36,37,38,39], it may be that some cases were ascribable to this zoonoses. Death from toxic diseases, concerning especially medical history and the symptoms reported (e.g., agitation, hypersalivation, tremors, seizures, convulsions, vomiting, diarrhea, etc.), may have been due to poisoning. Unfortunately, no specific tests were performed. As observed by other authors [31], generally the mortality for poisoning is prevalent in stray dogs rather than owned dogs. In the present study, a differentiation was not carried out. Undoubtedly, poisoning remains a serious cause of the mortality for dogs and cats. Relating to trauma, some dogs had injuries affecting the central nervous system (head trauma), or the abdomen and/or the thorax as a consequence of a vehicle accident. The cause of death directly related to intrathoracic and intra-abdominal injury was noted by Kolata and Johnston (1970) [40]. Although mechanical ventilation, fluid resuscitation, and vasopressors were utilized, animals died in the hospital from multiple organ failure or exsanguination with consequent cardiopulmonary arrest in the first 48 h after admission (acute death). Additionally, in human medicine, organ failure and death are usually one of the main causes of mortality among trauma patients [41]. The severity depends on the anatomic location of the injury, the nature of the traumatic pathologic process, and the organ or structure involved, as shown in previous studies [40].

Deaths by AD were also due to trauma and most often occurred within 1 h of being admitted. Information regarding the outcome was unavailable given the retrospective nature of this study. For example, one patient, as a result of spinal fractures, was euthanized due to a poor prognosis and another due to the financial restrictions of the owner to afford surgical intervention.

Neoplasia was the main cause of death by E, which may have potentially affected the dog’s quality of life and led the owner to decide to put down his dog, followed by degenerative (2nd rank) and congenital diseases (3rd rank). These outcomes are in accordance with other authors that reported as leading causes of euthanasia in dogs the presence of neoplastic, musculoskeletal, and neurological diseases [1,21]. The straight correlation between E and the presence of neoplasia reported in the present study may be strongly associated with the age of the dogs included. Indeed, neoplastic diseases, musculoskeletal disorders, and neurological disorders are considered the main causes of death in dogs over 3 years, while, more frequently, dogs under 3 years were submitted to E in the presence of behavioral disturbance, gastrointestinal diseases, and trauma [20]. Another aspect for consideration is that, unfortunately, in some neoplasia, the symptoms are often not recognized until it is too late, because many of them may be common to a large range of diseases and so a diagnosis of cancer cannot be made on symptoms alone. Furthermore, likely, hospitals generally provide more attractive medical services than clinics becoming more competitive to receive cancer patients who would potentially die in these veterinary structures [28].

The most frequent OS involved in deaths by UD was gastrointestinal due primarily to parvo-enteritis, gastric volvulus, and dilation. As reported in the medical records, dogs died when their stomach distended and rotated upon itself, compressing vital blood vessels and organs within the abdomen. Cardiovascular, respiratory, and renal/urinary systems followed in the 2nd, 3rd, and 4th rank, respectively. Cardiac diseases are the common cause of morbidity and mortality in dogs, representing around 8% of the whole mortality in dogs older than 10 years of age [42]. Additionally, they are considered a principal contributor to canine mortality after neoplastic diseases and trauma and before kidney diseases [42,43].

The patients in whom E was performed on were affected mainly by neurological, hepatic, and hematopoietic diseases. The onset of the clinical signs of these diseases often appears more evident and disabling than other diseases that have poor clinical results or a silent development, and therefore the decision-making process to perform E or not could be influenced by this aspect.

The main cause of death by AD was trauma and the OS involved was musculoskeletal. It is reported that in 9% of dog mortality, the dogs were initially referred for trauma [40]. The cause of injury, the amount of distribution of kinetic energy discharged into the animal, and the anatomic location of the injury are considered as factors that influences the outcome and the onset of mortality [40]. The education of owners on the management of the animal’s environmental hazards plays an important role in the onset of trauma.

Sex was identified in the current study as a risk factor. Previous research results on the correlation between the sterilization status and disease-specific risk of death have shown that generally neutered female dogs have a higher incidence to die of specific cancers (i.e., lymphoma and osteosarcoma) [44,45,46,47,48].

In this study, female gender was identified as a significant risk predictor of death by E. This finding is in line with previous studies [32] where authors have suggested that entire females lived shorter lives than neutered females, entire males, or neutered males [48]. This is in contrast with human medicine where studies clearly demonstrate a survival benefit among women compared with men and postmenopausal women, for example in blunt trauma injury, thanks to the protective effects of estrogen [49]. Nevertheless, larger prospective studies may be necessary as intact females outnumbered spayed females by nearly a ratio of 2:1 in this study group.

Conversely, in deaths by UD, male dogs are at a higher risk compared to female dogs. The rationale behind this finding is not completely clear because the specific causes of death (PP and/or OS) regarding the gender was not investigated. It is possible, however, that there is a direct cause-and-effect relationship between reproduction and the cause of death. Generally, the absence of gonadal hormones and altered feedback on the pituitary or adrenal hormonal axes could be considered significant factors [48]. Other studies have highlighted that sterilized dogs are less likely to die due to infection than intact dogs, due to reducing levels of progesterone and testosterone, recognized as immunosuppressive hormones [50,51,52].

Although the risk of all-cause mortality may be associated with the variable “age”, in the current study this was identified as a significant predictor of deaths both by E and UD. Old dogs are often affected by diseases such as cancer, cardiac failure, and renal (kidney) failure that cause a persistent and incurable inability to eat, vomiting, pain, distress or discomfort, or difficulty in breathing. Additionally, the expectation of life in an old dog is less than in a young dog.

Given that the overall median life expectancy in dogs is 12.0 years [25], it is conceivable that owners may lose numerous dogs during their lifetime [20], with consequent distress not related to whether the method of death was E or UD. Therefore, classifying the causes and risk factors of death in dogs may be considered an instrument advantageous to improving their health and welfare, consequently prolonging their life expectancy [27].

In this study, data stratified by age revealed that dogs ≥ 96 months old had the highest death rates. Age was also identified as a significant predictor of deaths both by E and UD in the current study. The increasing age is the most common risk factor for E for UD, as reported by other authors relating to euthanasia [53], probably because it is simpler to opt for euthanasia in older dogs than in younger dogs [4]. However, aging animals must also receive veterinary care and, where appropriate, palliative care [54].

The condition that exposes dogs to the greatest risk of death by E is the PP of neoplastic nature, but the different types of tumors were not specified in this study. These finding corroborate previous epidemiological studies on euthanasia of dogs [19].

While in UD, a significant risk predictor is infection/inflammatory conditions. This highest exposition in infectious diseases may be due to a lack of knowledge and the negligence of owners, who are affected by socio-economic aspects [55].

## 5. Limitations

The study poses some limitations due its retrospective nature and to the collection of data from a single veterinary hospital. Each hospital manages a dog population derived from its geographic area, thus a single hospital may not be representative of all other structures in Italy.

It also provides a low number of cases, making the results more prone to random error and limitations. The lists of clinical signs and disease diagnoses recorded in medical records were often limited and not exhaustive, as well as the variable breed. Several diagnoses reported in the study were based on what was noted by the veterinarian in the medical record and we were unable to verify whether these diagnoses had been confirmed by diagnostic testing.

Nevertheless, the data presented in this study can serve as a starting point help to design further studies of a wider population of dog.

## 6. Conclusions

In the last decades, we have observed an increasing interest in the research on mortality and longevity in dogs [1,24,25,30,56,57,58,59].

Making euthanasia decisions may be a moral dilemma for the veterinarian and a time of emotion turmoil for the owner [4,60]. In fact, some owner-related factors, such as the emotional factors, financial constraints, and not having the time to care for a sick dog, may further influence decision-making when it comes to euthanasia versus unassisted death.

When considering choosing euthanasia, decision-makers should prioritize the dog’s quality of life rather than the quantity. In guiding the decision to euthanize and when the animal is the patient, the veterinarian may inform the owner of the characteristics of the quality life which his/her animal will have in relation to the severity of the disease and the perception of the pain. The respect for animal welfare and dignity must be viewed as the main principle in the veterinary practice, with the consequent need for a better understanding of diseases’ course and animal pain management from an ethical viewpoint [9,10].

## Figures and Tables

**Figure 1 vetsci-09-00554-f001:**
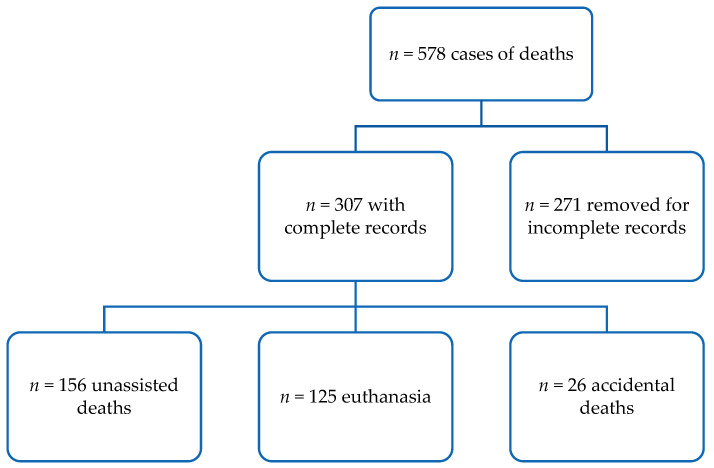
Flow chart showing death methods.

**Table 1 vetsci-09-00554-t001:** Percentages and ranks for categorical variables and *p*-values for comparison between E and UD.

Variable	Method of Death							
**PP** ** ^(5)^ **	**E** ** ^(1)^ **		**UD** ** ^(2)^ **			**AD** ** ^(3)^ **	**Total Deaths** ** ^(4)^ **	
	**(%)**	**Rank**	**(%)**	**Rank**	** *p* **	**(%)**	**(%)**	**Rank**
Congenital	60	3	40	5	0.0009	0	1.6	7
Degenerative	64.3	2	35.7	6	<0.0001	0	9.1	4
Inflammatory/Infectious	34.9	6	65.1	2	<0.0001	0	49.5	1
Metabolic	0	/	100	1	<0.0001	0	1.3	6
Neoplastic	75.6	1	24.4	8	<0.0001	0	14.37	3
Toxic	38.5	5	61.5	3	<0.0001	0	4.2	5
Traumatic	18.2	7	34.5	7	0.0023	47.3	17.9	2
Vascular	40	4	60	4	0.0009	0	1.6	7
Total							307	
**OS ^(6)^**	**(%)**	**Rank**	**(%)**	**Rank**	** *p* **	**(%)**	**(%)**	**Rank**
Cardiovascular	31.2	9	68.8	2	<0.0001	0	5.2	6
Dermatological	50	4	50	5	0.999	0	2	9
Gastrointestinal	12.7	10	87.3	1	<0.0001	0	20	2
Hematopoietic	62.5	3	37.5	7	<0.0001	0	2.6	8
Hepatic	75	2	25	8	<0.0001	0	1.3	10
Musculoskeletal	46.7	5	20	9	<0.0001	33.3	4.9	7
Neurological	75.6	1	17.1	10	<0.0001	7.3	13.4	4
Renal/urinary	45.8	6	54.2	4	0.1624	0	15.6	3
Respiratory	45.2	7	54.8	3	0.1103	0	10.1	5
Systemic/multiorgan	36	8	40	6	0.4937	24	24.4	1
Total							307	

^(1)^ E = euthanasia; ^(2)^ UD = unassisted death; ^(3)^ AD = accidental death; ^(4)^ total deaths = E + UD + AD; ^(5)^ PP= physio-pathological process; ^(6)^ OS= organ system.

**Table 2 vetsci-09-00554-t002:** Results of univariate and multivariate logistic regression analysis for gender, purebred, age at death, sterilization, owned, and PPs as risk factors associated with “Euthanasia yes or no”.

Covariates	Univariate			Multivariate		
	OR	95% I.C.	*p*-Value	OR	95% I.C.	*p*-Value
Gender (F vs. M)	1.954	1.231–3.101	0.005	1.867	1.075–3.243	0.027
Purebred (yes vs. no)	1.071	0.654–1.752	0.785	1.131	0.608–2.101	0.698
Age at death (months)	1.010	1.006–1.014	0.001	1.009	1.004–1.013	0.001
Sterilization (yes vs. no)	1.570	0.854–2.885	0.146	1.403	0.685–2.873	0.354
Owned (yes vs. no)	0.978	0.607–1.574	0.607	1.713	0.895–3.280	0.104
Infectious/Inflammatory conditions (yes vs. no)	0.617	0.390–0.977	0.039	0.481	0.238–0.971	0.041
Neoplastic diseases (yes vs. no)	5.808	2.811–12.003	0.001	2.585	1.033–6.469	0.043
Traumatic events(yes vs. no)	0.265	0.128–0.549	0.002	0.230	0.089–0.592	0.002
Degenerative	2.893	1.287–6.503	0.010	4.928	1.805–13.45	0.002
Toxic	0.906	0.289–2.837	0.866	1.358	0.221–8.352	0.742

**Table 3 vetsci-09-00554-t003:** Results of univariate and multivariate logistic regression analysis for gender, purebred, age at death, sterilization, owned and PPs as risk factors associated with “Unassisted death (yes or no)”.

Covariates	Univariate			Multivariate		
	OR	95% I.C.	*p*-Value	OR	95% I.C.	*p*-Value
Gender (M vs. F)	2.066	1.308–3.265	0.002	1.878	1.096–3.218	0.022
Purebred (yes vs. no)	0.987	0.609–1.599	0.957	1.165	0.637–2.130	0.620
Age at death (months)	1.937	1.037–3.619	0.003	1.743	1.028–2.956	0.013
Sterilization (yes vs. no)	0.721	0.392–1.327	0.293	1.049	0.511–2.152	0.896
Owned (yes vs. no)	0.736	0.460–1.176	0.199	0.767	0.473–1.186	0.102
Infectious/Inflammatory conditions (yes vs. no)	3.212	2.014–5.122	0.001	2.179	1.079–4.400	0.030
Neoplastic diseases (yes vs. no)	0.261	0.127–0.537	0.001	0.403	0.161–1.004	0.055
Traumatic events (yes vs. no)	0.443	0.241–0.814	0.009	0.383	0.159–0.925	0.033
Degenerative	0.506	0.226–1.135	0.099	0.293	0.057–1.500	0.141
Toxic	1.578	0.505–4.938	0.433	0.726	0.118–4.480	0.731

## Data Availability

For other information contact the corresponding author

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
