# Peer review of "Risk Factors Regarding Dog Euthanasia and Causes of Death at a Veterinary Teaching Hospital in Italy: Preliminary Results"

_vetsci, 2022, doi:10.3390/vetsci9100554_

Round 1
Reviewer 1 Report
Dear Authors,
I went through this paper with great interest and motivation, being the topic of this paper of extreme relevance, but I came across fundamental issues mainly related to the terminology used and the quality of the text.
The second sentence of the abstract is unclear if not incorrect (This study investigates risk factors associated with the main causes 11 and methods of death (euthanasia or unassisted) in a population of dogs). In veterinary medicine euthanasia is a method of killing an animal, not a method of death.
One of the aim of the study is to “identify the risk factors associated with the main causes and method of death (unassisted or euthanasia). What do you mean with “unassisted method of death”? My understanding of the adjective “unassisted” is “not helped by anyone or anything”. Perhaps you refer to “natural causes of death” but then they should not use the term “type of deaths” (Line 105) and “death methods” (Figure 1).
Line 20 of the Abstract reads, “The main causes of death for euthanasia were inflammatory or infectious diseases (49.5%), trauma disorders (17.9%%), and neoplasia (14.7%)”. However, you will notice a mistake going to Line 110, as these percentages actually refer to the most frequent causes of death, namely inflammatory/infectious processes (49.5%), traumatic events (17.9%) and neoplasia (14.37%), whilst Line 111 reads “but E was mainly carried 111 out in neoplastic (34;75.6%), degenerative (18; 64.3%) and congenital (3; 60%) diseases”.
Please double check thoroughly data, and terminology, consistency and coherence.
Please clarify the following sentences:
- (Line 240 – 243) "Patients dead by E due mainly to neurological, hepatic, and hematopoietic diseases. It is possible to relate this data to a large showing of clinical findings related to these diseases that are more evident, influencing the decision-making process, conversely to other diseases that present poor clinical findings or silent development”
- (Line 248) “The education of pet owners on pet’s management of pet at environmental hazards play an important role in the onset of trauma”
- (Line 270) “Ultimately, the differences vary heavier concerning neutering status”.
Please make an effort to produce a text of a better quality.
In addition:
-
The sample size appears to be not representative for an average volume of activity of a veterinary hospital, only 307 dogs assisted in 20 years (1,2/month) met the eligibility criteria (which ones?). Perhaps the number of casualties were largely underreported;
-
Conclusions are weak and not always supported by the results. The very last statement sounds extremely nice but there are very few elements to understand why and how “this study may be used to evaluate strategic interventions and health promotion strategies”.
Author Response
Dear Reviewer,
We would like to thank the reviewer for her/his time and for all the valuable suggestions. Here in you find a revised version the manuscript according to the reviewers’ comments. The changes made in the manuscript to address comments are written in red.
- The second sentence of the abstract is unclear if not incorrect (This study investigates risk factors associated with the main causes and methods of death (euthanasia or unassisted) in a population of dogs). In veterinary medicine euthanasia is a method of killing an animal, not a method of death.
A: Certainly, we agree that in veterinary medicine euthanasia is a method of killing. Some laws regulate the kill methods used in approved slaughterhouses, and they are related to food animals. There are laws for humane killing that is used i.e. for scientific purposes. Yet, animal killing could be considered a crime if it is carried out without necessity and with cruelty. However, underlined these concepts, euthanasia and unassisted death can be considered methods of death as also reported by other Authors (Pegram et al., 2021). In this article, the Authors categorize methods of death as euthanasia or unassisted.
- One of the aim of the study is to “identify the risk factors associated with the main causes and method of death (unassisted or euthanasia). What do you mean with “unassisted method of death”? My understanding of the adjective “unassisted” is “not helped by anyone or anything”. Perhaps you refer to “natural causes of death” but then they should not use the term “type of deaths” (Line 105) and “death methods” (Figure 1).
A: I’m sorry about this misunderstanding. Unassisted death, as reported by Yu et al. (2021) means: when the patient dies upon presenting to the hospital or the animal dies whilst diagnosis and/or treatment was attempted/ confirmed. However, in line 155 and table 1 we have replaced the word «type» with «method».
- Line 20 of the Abstract reads, “The main causes of death for euthanasia were inflammatory or infectious diseases (49.5%), trauma disorders (17.9%%), and neoplasia (14.7%)”. However, you will notice a mistake going to Line 110, as these percentages actually refer to the most frequent causes of death, namely inflammatory/infectious processes (49.5%), traumatic events (17.9%) and neoplasia (14.37%), whilst Line 111 reads “but E was mainly carried 111 out in neoplastic (34;75.6%), degenerative (18; 64.3%) and congenital (3; 60%) diseases”.
A: Thank you very much for your observation. It is correct and we have modified the sentence in the abstract.
Please clarify the following sentences:
- (Line 240 – 243) "Patients dead by E due mainly to neurological, hepatic, and hematopoietic diseases. It is possible to relate this data to a large showing of clinical findings related to these diseases that are more evident, influencing the decision-making process, conversely to other diseases that present poor clinical findings or silent development”
A : The sentence has been rephrased.
- (Line 248) “The education of pet owners on pet’s management of pet at environmental hazards play an important role in the onset of trauma”
A: One of the factors that may influence an animal's chances of being injured is the owner's management of the animal's environment. The veterinarian's role in dealing with trauma should be also educating pet owners about the common environmental hazards that all too frequently affect their pets.
We have modified the sentence.
- (Line 270) “Ultimately, the differences vary heavier concerning neutering status”.
A: We have deleted the sentence.
- The sample size appears to be not representative for an average volume of activity of aveterinary hospital, only 307 dogs assisted in 20 years (1,2/month) met the eligibility criteria (which ones?). Perhaps the number of casualties were largely underreported;
A: 307 are not dogs assisted in this referral veterinary hospital, but the number of animals died.
- Conclusions are weak and not always supported by the results. The very last statement sounds extremely nice but there are very few elements to understand why and how “this study may be used to evaluate strategic interventions and health promotion strategies”.
A: Identifying risk factors helps to prevent problems from occurring. Some neoplasia, such as mammary cancer, merit serious consideration as a core principle of any wellness program intended to optimize overall longevity and/or to carry out correct euthanasia decision-making. Therefore, for example, an intervention could be education and information programs in order to contribute to creating awareness about euthanasia and its choice.
Kind Regards
Prof. Annamaria Passantino

Reviewer 2 Report
Dear authors,
Thank you very much for the interesting read of your submitted manuscript on “Risk factors regarding dog euthanasia and causes of death at a Veterinary Teaching Hospital” that focusses on the investigation of risk factors associated with the main causes and methods of death. The manuscript covers an important topic in veterinary practice by comparing euthanasia and unassisted death of a sample of 307 dogs. However, there are some issues and flaws that need to be considered in the re-writing process. Please find attached a document including recommended changes for improvement.

Author Response
Dear Reviewer,
We would like to thank the reviewer for her/his time and for all the valuable suggestions.
Herein you find a revised version the manuscript according to the reviewers’ comments. The changes made in the manuscript to address comments are written in red.
Abstract:
Even though the abstract displays main results of the study, at its current stage it can be
improved in the structure to guide the reader in a better way through your idea, methods,
results as well as conclusions of your study.
A: The abstract has been improved.
L 10: Why is “the” in bold letters?
A: We have corrected it.
L 10-11: The first sentence does not guide the reader very well to the intention of the study. I
would unpack the first sentence in order to get a better idea what is meant by risk factors.
Further, you should specify that you focus on “causes” that are based on medical factors
(specific diseases etc.) only.
A: We have specified the typology of risk factors and causes.
L 12: “… in a population of dogs”. Please add the number of dogs included in your study.
A: We deem it appropriate not to specify the number of dogs in this line. It is specified in brackets.
L 13: Why do you write authors with capital letter?
A: We have corrected it.
L 17-20: The description of what regression analyses you used are distracting in an abstract.
A: We believe it is not. We prefer to leave it.
Introduction:
In general, the introduction needs to be improved. At its current state it does not guide the
reader to the addressed aim and research questions of the study in a sufficient way.
A: We have revised the introduction based on your suggestions and the other reviewer.
L 35-39: This paragraph needs to be unpacked. When it comes to decision-making in the
context of euthanasia, these situations are often complex and influenced by diverse factors. I
think it is important to introduce the topic of euthanasia and challenges related to decisionmaking
in a more thoughtful way including different situations. Based on this, you can
highlight that your study focusses on medical reasons. At the moment you mention in one
sentence “rationally made” euthanasia and then address in the next sentence the medical
reasons (L: 47-53).
A: We have argued the topic related to decision making.
Keeping this in mind, the authors should spend some time and effort to come up with a clear
story line for their study by introducing relevant aspects related to euthanasia versus
unassisted death. None of your investigated risk factors is introduced in your introduction that
guides the reader to your investigation. Hence, the introduction needs to be heavily rewritten
by contextualizing the investigated aspects of the study and demonstrating that the authors
have the sufficient background knowledge based on existing literature.
A: We have rewritten the introduction
Material and Methods:
Section 2.1.:
L 70-77: Maybe consider for result section?
A: We have preferred to delete this part in materials and methods, as reported by other Authors.
Do you have an ethical approval for this study?
A: The Ethics review board of our Department have considered that the study cannot legally be categorized as an animal experiment given that data are recorded retrospectively as anonymized clinical data from the electronic database, therefore it did not require ethical approval. We have sent the letter of Ethics review board to Editor of the Journal.
L 76-77: Please specify how many dogs were excluded
A: Done in results.
Section 2.2:
Please find below some suggestions for additional analyses that would improve and increase
the impact of the manuscript (Section 3.2)
Results:
L 95: Do not start a sentence with a number
A: We have deleted the number at the beginning of the sentence.
L 97: display age of dogs in months and years.
A: For a more correct statistical evaluation, the reference period chosen for the collection and calculation of statistical results was in months and not years. Therefore, it is not possible to satisfy your request for modification.
L 98: Please use median values and then IQR instead of Q1-Q3.
A: Done in results
L 96: It is important to present the variable “breed” and include it as a predictor in your
regression analyses. However, against the background that some breeds are more likely to
suffer from cancer or other specific disease, I wonder why the authors do not present and use
the variable breed in a more detailed manner.
A: We didn’t include the variable breed because often in medical records it was not specifically detailed.
2. Rate and type of death
Figure 1: Not very informative as it is and just repetitive with the text.
Why not using a table and further differentiate between the three decades (1990-2020) you
included in you study. I think it would be really worthwhile and important to consider the
different decades which was totally neglected.
Recent developments in the field of veterinary medicine affect and influence patient care
including decision-making in the context of euthanasia. It can be hypothesized that there exist
differences between the three decades included. Hence, I would encourage the authors to
split the data set and present descriptive statistics by decade. Further, by conducting bivariate
statistics you could investigate whether differences exist between the three decades.
A: Figure 1 was replaced by a new figure.
We are sorry but unfortunately, we made a mistake. In the initial version of our manuscript, we considered data from 1990 to 2020; but the data referring to the first two decades were not collected with detailed information and the database was lacking due to the absence of numerous variables. Data from 2010 onwards were collected in detail. Therefore, we have decided, in the final version of the article, to consider the data referring to the last decade as they are more detailed. For this reason, we are unable to carry out what you requested with reference to the stratification by decade. We have proceeded to rectify the data collection period in the article. We apologize for this mistake.
3. Risk factors
Table 1: Gives a good overview. However, I would further conduct bivariate statistics in order
to identify differences between E and UD in the distribution of PP and OS listed in Table 1.
A: In Table 1 we reported the P-values obtained from the comparisons between "Euthanasia” and “Unassisted death” in relation to PP and OS.
Table 2 and 3: Please state (preferably in the methods section) why not include congenital or
vascular in your regression analyses. On what basis did you make your choice of predictor
variables?
A: In the method section we explained why congenital or vascular processes were not included in the regression analyses.
Discussion:
- In comparison to the introduction, the discussion is profound and the authors contextualize
their results in an interesting way. However, as I stated before, I recommend that the authors
conduct some further analyses by considering that they included a sample of dogs over a
period of 30 years. If you leave it like this, this constitute a main limitation of the study since
it neglects the fact that medicine is not static and hence influence decisions – especially when
it comes to end-of-life decisions. This limitation can easily be resolved by using it as a variable
of investigation.
A: Sorry for the mistake, as above said. The period considered is of ten years (2010-2020) and, for this reason, we didn’t conduct further analyses.
- When it comes to limitations of the study, in the re-vised version of your manuscript I would
expect a paragraph that addresses important limitations of your study and conducted analyses
(including your used predictor variables, e.g. see my comment concerning your variable
breed).
A: We have added a paragraph relating to the limitations
- Further, although the authors focus on medical causes of death only, it is important
to note that further aspects related to the vet (e.g. different assessment of cases and hence
different handling of dogs with respect of how to end life) and owner (e.g. emotional factors,
financial factors, not time to care for a sick dog) further influence decision-making when it comes to euthanasia versus unassisted death.
A: We have added these aspects in the conclusions.
Kind Regards
Prof. Annamaria Passantino

Reviewer 3 Report
Unfortunately my review is negative as this paper ultimately does not answer the questions it seeks to, and I believe the methodology is flawed.
The authors propose identifying potential risk factors for euthanasia as opposed to unassisted or accidental death looking at data from a single teaching hospital over a 30 year period. I think this notion has some value, although larger studies have already been performed, so unfortunately this study does not add anything new to existing literature.
There seems a lack of familiarity with literature about euthanasia, despite a long list of references.
The introduction is quite poor. I am not sure I agree with the first paragraph – the ability of veterinary treatment to prolong animal life, and the normalisation of euthanasia, are major drivers around current discussions of veterinary euthanasia, as are concerns about potential moral distress faced by veterinary team members regarding euthanasia (see for example Batchelor and McKeegan 2012, Kipperman 2018, Moses 2018, Quain 2021).
In the second paragraph you mention that medical decision making often contributes – I think this discussion needs to be expanded. How does it contribute? I would suggest that in many cases, veterinarians may recommend euthanasia based on poor quality of life. This is not a bad thing and in fact may be positive for the welfare of animals, depending on how you define welfare. You mention drugs that may shorten life – please provide examples.
Owners may not ask for euthanasia, in fact, they may seek to avoid it altogether – and client euthanasia refusal is another known source of moral distress among veterinary team members. Euthanasia may be recommended by a veterinarian, for example due to poor or deteriorating quality of life that cannot be restored through veterinary care. To what extent does the burden of patient care play a role?
You talk about euthanasia being “acceptable and necessary” – according to whom? It is useful and relevant to cite laws, for example laws regarding veterinary euthanasia in Italy. Are laws re euthanasia in Italy different, for example, to countries where studies on the incidence of euthanasia have been performed? If so, how?
On the second page, after the first paragraph, you don’t mention the potential for moral distress of veterinarians or animal owners themselves.
I am concerned about the methodology. How were data anonymised? Or were they deidentified after selection? It seems they were retrospectively accessed, but little detail is provided.
The time frame chosen is 30 years which is a large time frame, yet in this time at a teaching hospital, only 307 deaths were recorded? This is very surprising – the number seems very low. Were there issues accessing records? Was the data accessed complete for all of those years? I don’t feel there is adequate information re inclusion or exclusion criteria. How many dogs had an unknown cause of death? How did you differentiate these from UD or AD?
How did you distinguish UD from AD?
Re statistical analysis, was infectious or inflammatory conditions also “(yes/no)” (line 86)? What do you mean by toxic process? Would sepsis count and if so, would this count as toxics or infectious/inflammatory?
Also, given no specific tests were performed, how were diseases classified as “toxics”?
Less than 2/3 of the dogs in the cohort were owned. This seems unusual given this is a referral hospital – did it also function as a shelter? Was it that owners could not be immediately identified at the time dogs were admitted?
I’m not sure why covariates with a P value of >0.05 were included in the multivariate analysis? For example, if degenerative disease was not significant in the UV, why was it included in the MV and how do you interpret the significant result?
Re the “protective effect” of infection or inflammatory conditions against euthanasia – isn’t this likely because they caused death before euthanasia could occur? The same with trauma? I am confused as to why trauma had lower UD – was it because it had higher association with AD, or E?
I don’t really understand the lines 159-161 - what are you saying here?
Paragraph beginning Line 163 – there is also a study from New Zealand – Gardiner et al?
You mention 167 but you cannot calculate the mortality rate in this study as you don’t provide an overall population figure.
Re line 173, I am not sure this is a particularly useful hypothesis. We don’t know for example whether this hospital sees animals for “wellness” examinations. Apart from such, isn’t it clear that people would not present unwell animals unless clinical signs were apparent?
You mention presumptive dx but then you classify deaths according to underlying pathophysiological processes – yet not enough information is provided in the methodology to give the reader any level of confidence in this classification. This is a major methodological oversight.
Line 187 – or it might indicate that owners could not afford to pay for veterinary care.
Re Lines 196-199 – these signs are not pathognomic for toxicity and not a basis to determine cause of death alone. As you said, no tests were performed. The statement that death from toxic diseases may be caused by poisoning is true. You make a comment about stray dogs being more likely to be poisoned – so were stray dogs more likely to present to this facility?
Re Line 213-214 – should this not be in the methodology in inclusion criteria for AD? How were these deaths differentiated from UD? Failure to differentiate these categories is a major methodological weakness of this study.
Lines 215-217 – when you say only a few dogs, how many? What you seem to be suggesting is that the majority of dogs that were euthanased were not euthanased because they had a poor prognosis. I find this very surprising.
Line 228 – 229 – I think this is a generalisation that requires critical assessment. Neoplasia is a broad category of disease. Some neoplastic disease eg skin tumours may be highly visible and easily detected. I don’t think this is a helpful statement.
Line 229-231 – I am really not sure what you mean by this sentence.
Line 249 – do you mean prevention?
Line 271 – might this reflect an owner’s willingness to invest in the care of their animal?
In the second paragraph on page 8 you talk about prolonging the life expectancy of animals, but make no comment about welfare or quality of life. Yet in the conclusion you mention “it should be considered giving the prioritisation to the quality of life rather than the quantity”. This has the effect of undermining your conclusion.
Line 281-282 – could it not be argued that the risk of all-cause mortality increases with age? Surely this is not unexpected?
Why is it “simpler” to euthanase older dogs than younger?
There is no limitations section in the body of the text. There are known limitations associated with retrospective data sets, as well as the fact (mentioned only in the conclusion) that the data is from a single institution. The sample size is small.
I don’t see how these findings may help prevent unnecessary euthanasia. The authors provide no explanation as to how these findings can prevent unnecessary euthanasia.
Minor changes:
No need to capitalise “Authors” (done throughout manuscript)
Trauma disorders/traumatic events should be replaced in text with “trauma”
Line 48 – avoid “put animal down” – use euthanase
Line 91: P <0.05 (not 0,05)
Line 274: delete several. Do you mean severe distress?
Author Response
Dear Reviewer,
We would like to thank the reviewer for her/his time and for all the valuable suggestions.
Herein you find a revised version the manuscript according to the reviewers’ comments.
The changes made in the manuscript to address comments are written in red.
The introduction is quite poor.
A: We have revised the introduction.
I am not sure I agree with the first paragraph – the ability of veterinary treatment to prolong animal life, and the normalisation of euthanasia, are major drivers around current discussions of veterinary euthanasia, as are concerns about potential moral distress faced by veterinary team members regarding euthanasia (see for example Batchelor and McKeegan 2012, Kipperman 2018, Moses 2018, Quain 2021).
A: We have deleted the sentence of the first paragraph, and we have added sentences relating to moral distress faced by veterinarians.
In the second paragraph you mention that medical decision making often contributes – I think this discussion needs to be expanded. How does it contribute? I would suggest that in many cases, veterinarians may recommend euthanasia based on poor quality of life. This is not a bad thing and in fact may be positive for the welfare of animals, depending on how you define welfare. You mention drugs that may shorten life – please provide examples.
A: We have discussed these considerations.
Owners may not ask for euthanasia, in fact, they may seek to avoid it altogether – and client euthanasia refusal is another known source of moral distress among veterinary team members. Euthanasia may be recommended by a veterinarian, for example due to poor or deteriorating quality of life that cannot be restored through veterinary care. To what extent does the burden of patient care play a role?
You talk about euthanasia being “acceptable and necessary” – according to whom? It is useful and relevant to cite laws, for example laws regarding veterinary euthanasia in Italy. Are laws re euthanasia in Italy different, for example, to countries where studies on the incidence of euthanasia have been performed? If so, how?
A: We have revised this part of the introduction
On the second page, after the first paragraph, you don’t mention the potential for moral distress of veterinarians or animal owners themselves.
A: We have mentioned it in the introduction
I am concerned about the methodology. How were data anonymised? Or were they deidentified after selection? It seems they were retrospectively accessed, but little detail is provided.
A: Data were anonymized after selection. We have specified it in the text.
The time frame chosen is 30 years which is a large time frame, yet in this time at a teaching hospital, only 307 deaths were recorded? This is very surprising – the number seems very low. Were there issues accessing records? Was the data accessed complete for all of those years? I don’t feel there is adequate information re inclusion or exclusion criteria. How many dogs had an unknown cause of death? How did you differentiate these from UD or AD?
A: Thank you very much for this observation. Although in lines 120-121 we have reported that dogs for which the sole diagnosis was “death due to unknown’’ were excluded from the analysis and in paragraph 3.1. Descriptive statistics of the study population we have reported that in total, 307 dogs met the eligibility criteria (line 145), we didn’t report the number of these deaths. We have reformulated the sentence (lines 144-145) and changed the figure 1.
How did you distinguish UD from AD?
A: We have specified it in introduction and in materials and methods.
Re statistical analysis, was infectious or inflammatory conditions also “(yes/no)” (line 86)? What do you mean by toxic process? Would sepsis count and if so, would this count as toxics or infectious/inflammatory?
A: Yes, or inflammatory conditions also “(yes/no)”. For toxic process we mean having to do with poison (see lines 252-254)
Also, given no specific tests were performed, how were diseases classified as “toxics”?
A: We have classified the process on the basis of the history and symptoms (in all dogs it was not possible to carry out specific tests)
Less than 2/3 of the dogs in the cohort were owned. This seems unusual given this is a referral hospital – did it also function as a shelter? Was it that owners could not be immediately identified at the time dogs were admitted?
A: In this hospital stray dogs were also admitted in relation to a convention with municipal.
I’m not sure why covariates with a P value of >0.05 were included in the multivariate analysis? For example, if degenerative disease was not significant in the UV, why was it included in the MV and how do you interpret the significant result?
A: Our choice to insert all variables in the construction of the multiple regression had the purpose of explaining the variation of the dependent variable for each different predictive variable, present singularly in the model, and for the complex of interactions of the independent variables.
Re the “protective effect” of infection or inflammatory conditions against euthanasia – isn’t this likely because they caused death before euthanasia could occur? The same with trauma? I am confused as to why trauma had lower UD – was it because it had higher association with AD, or E?
I don’t really understand the lines 159-161 - what are you saying here?
A: Thank you for your suggestion. The sentence has been rephrased to clarify the opinion reported.
Paragraph beginning Line 163 – there is also a study from New Zealand – Gardiner et al?
A: Yes, there is another preliminary study of Gates et al. (2017). We have added it.
You mention 167 but you cannot calculate the mortality rate in this study as you don’t provide an overall population figure.
A: The sentence has been modified.
Re line 173, I am not sure this is a particularly useful hypothesis. We don’t know for example whether this hospital sees animals for “wellness” examinations. Apart from such, isn’t it clear that people would not present unwell animals unless clinical signs were apparent?
A: We have deleted this sentence
You mention presumptive dx but then you classify deaths according to underlying pathophysiological processes – yet not enough information is provided in the methodology to give the reader any level of confidence in this classification. This is a major methodological oversight.
Data were collected retrospectively, and the outcome classified as dead (unassisted and/or accidental) or euthanized. Also, we are reported physio-pathological process (PP) affecting the animals at the moment of diagnosis.
Line 187 – or it might indicate that owners could not afford to pay for veterinary care.
A: Surely, I added this concept.
Re Lines 196-199 – these signs are not pathognomic for toxicity and not a basis to determine cause of death alone. As you said, no tests were performed. The statement that death from toxic diseases may be caused by poisoning is true. You make a comment about stray dogs being more likely to be poisoned – so were stray dogs more likely to present to this facility?
A: Signs of poisoning in dogs may vary depending on the type of poison (from gastrointestinal signs - as vomiting, diarrhea, extreme salivation, loss of appetite – to kidney or liver failure). However, we have suspected the death for toxic diseases on the basis of the history and symptoms.
In this hospital were also admitted stray dogs (35.5%).
Re Line 213-214 – should this not be in the methodology in inclusion criteria for AD? How were these deaths differentiated from UD? Failure to differentiate these categories is a major methodological weakness of this study.
A: AD were due to road traffic accidents and other accidents. We have specified it in the text (line 111)
Lines 215-217 – when you say only a few dogs, how many? What you seem to be suggesting is that the majority of dogs that were euthanased were not euthanased because they had a poor prognosis. I find this very surprising.
A: We have rephrased the sentence.
Line 228 – 229 – I think this is a generalisation that requires critical assessment. Neoplasia is a broad category of disease. Some neoplastic disease eg skin tumours may be highly visible and easily detected. I don’t think this is a helpful statement.
A: We have modified the sentence
Line 229-231 – I am really not sure what you mean by this sentence.
A: We have revised the sentence.
Line 249 – do you mean prevention?
A: Yes. We have modified the sentence
Line 271 – might this reflect an owner’s willingness to invest in the care of their animal?
A: Although recent studies have shown that for some dog breeds spaying may be associated with an increased risk of debilitating joint disorders and certain cancers, in our view the decision to spay the animal cannot be related to willingness in this context.
In the second paragraph on page 8 you talk about prolonging the life expectancy of animals, but make no comment about welfare or quality of life. Yet in the conclusion you mention “it should be considered giving the prioritisation to the quality of life rather than the quantity”. This has the effect of undermining your conclusion.
A: We have deleted the sentence relating to prolonging the life expectancy of animals and added a sentence in the conclusions.
Line 281-282 – could it not be argued that the risk of all-cause mortality increases with age? Surely this is not unexpected?
A: The sentence has been rephrased in accordance with your suggestion.
Why is it “simpler” to euthanase older dogs than younger?
A: Old dogs are often affected by diseases such as cancer, cardiac failure, and renal (kidney) failure that cause persistent and incurable inability to eat, vomiting, pain, distress or discomfort, or difficulty in breathing. Also, the expectation of life in old dog is less than a young.
There is no limitations section in the body of the text. There are known limitations associated with retrospective data sets, as well as the fact (mentioned only in the conclusion) that the data is from a single institution. The sample size is small.
A: We have added another paragraph (5. Limitations)
I don’t see how these findings may help prevent unnecessary euthanasia. The authors provide no explanation as to how these findings can prevent unnecessary euthanasia.
A: The sentence has been deleted.
Minor changes:
No need to capitalise “Authors” (done throughout manuscript). A: Another reviewer asked us to capitalize Authors
Trauma disorders/traumatic events should be replaced in text with “trauma”
A: Done
Line 48 – avoid “put animal down” – use euthanase.
A: Done
Line 91: P <0.05 (not 0,05)
A: Done
Line 274: delete several. Do you mean severe distress?
A: Done.
Kind Regards
Prof. Annamaria Passantino

Round 2
Reviewer 2 Report
Thanks for sending me the revised version of your manuscript which can be accepted in its current form.
Author Response
Dear Reviewer,
We would like to thank you for your time.
Best regards,
Annamaria